

# Cross-cultural adaptation and psychometric properties of the Arabic version of the Central Sensitization Inventory in people with chronic musculoskeletal pain

Sarah E. Tamboosi[1], Hosam Alzahrani[2], Fahad H. Alshehri[2], Msaad Alzhrani[3] and Yasir S. Alshehri[4]

[1] Department of Physical Therapy, Al-khorma General Hospital, Taif, Saudi Arabia
[2] Department of Physical Therapy, College of Applied Medical Sciences, Taif University, Taif, Saudi Arabia
[3] Department of Physical Therapy and Health Rehabilitation, Majmaah University, Majmaah, Saudi Arabia
[4] Department of Physical Therapy, College of Medical Rehabilitation Sciences, Taibah University, Madinah, Saudi Arabia

Corresponding author
Hosam Alzahrani,
halzahrani@tu.edu.sa

## ABSTRACT

**Background:** The Central Sensitization Inventory (CSI) is a patient-reported screening instrument that can be used to identify and assess central sensitization (CS)/Central Sensitization Syndrome (CSS)-related symptoms.

**Objective:** The aim was to translate the CSI into Arabic (CSI-Ar) and to subsequently validate its psychometric properties.

**Design:** Cross-sectional.

**Methods:** The CSI was translated and cross-culturally adapted into Arabic, and validated following international standardized guidelines. This study included patients with chronic musculoskeletal pain ($n = 264$) and healthy control participants ($n = 56$). Patients completed the CSI-Ar, Pain Catastrophizing Scale (PCS), Depression, Anxiety, and Stress scale (DASS-21), Tampa Scale of Kinesiophobia (TSK), and 5-level EuroQol-5D (EQ-5D). Patients completed the CSI-Ar twice to assess test–retest reliability. To evaluate discriminative validity, healthy controls participants completed the CSI-Ar. Statistical analyses were conducted to test the internal consistency, reliability, and structural, construct and discriminant validity of CSI-Ar.

**Results:** The CSI-Ar showed acceptable internal consistency (Cronbach's alpha = 0.919) and excellent test–retest reliability (intraclass correlation coefficient = 0.874). The CSI-Ar scale had significant correlations ($P < 0.001$) with all PCS subscales and total score (Spearman's rho = 0.459–0.563, $P < 0.001$), all DASS-21 subscales and total score (Spearman's rho = 0.599–0.685, $P < 0.001$), the TSK (Spearman's rho = 0.395, $P < 0.001$), and the EQ-5D (Spearman's rho = −0.396, $P < 0.001$). The Mann-Whitney U-test showed a statistically significant difference between the patient group and the healthy control group ($P < 0.001$), with the healthy controls displaying a lower average CSI-Ar score ($12.27 \pm 11.50$) when compared to

the patient group (27.97 ± 16.08). Factor analysis indicated that the CSI-Ar is a unidimensional tool.

**Conclusion:** The CSI-Ar is a reliable and valid screening tool that can be used to assess CS/CSS-related symptoms in Arabic-speaking people with chronic musculoskeletal pain.

## INTRODUCTION

Chronic musculoskeletal pain is a leading cause of disability worldwide (*Vos et al., 2017*). According to the Global Burden Disease (GBD) statistics, 1.75 billion individuals worldwide suffer from chronic musculoskeletal pain (*Cieza et al., 2020*). As a result of chronic musculoskeletal (MSK) pain, daily tasks and activities become more difficult, medicines are used more frequently, and there is a larger possibility of sick leave and disability pensions, which in turn have a negative impact on quality of life. Furthermore, it is a major public health concern, resulting in substantial costs for healthcare systems and disability insurance (*Cimmino, Ferrone & Cutolo, 2011*).

A number of studies have focused on the phenomenon of central nervous system hypersensitivity in chronic pain patients (*Woolf, 2011*). This phenomenon is known as central sensitization (CS). The CS is a neurophysiological disorder causes hyperexcitability of the central nervous system. According to Woolf, the CS is "operationally defined as an increase in neural signaling in the central nervous system that causes hypersensitivity to pain" (*Woolf, 2011*). The CS is indicated for a variety of chronic pain disorders, including fibromyalgia (*Vierck, 2006*), whiplash (*Curatolo et al., 2001*), low back pain (*Roussel et al., 2013*), and osteoarthritis (*Lluch Girbés et al., 2013*).

*Yunus (2007)* used the term "Central Sensitization Syndrome (CSS)" to describe a chronic disease in which CS appears to be a common cause. The author proposed renaming these disorders to (CSS) and introduced the idea that CS may be a common trait that causes similar overlapping symptoms in these syndromes. In addition to the absence of structural pathology, the majority of CSS share objectively a lower pain threshold and heightened pain sensitivity (*Aaron & Buchwald, 2001*), which is an important feature of the CS (*Latremoliere & Woolf, 2009*).

In the past few decades, a complete clinical patient-centered biopsychosocial assessment and therapy approach has been evacuated for this complex patient population (*Nijs et al., 2014*; *Wijma et al., 2016*). The ability to recognize when presenting symptoms are related to CS can help clinicians choose the most relevant and effective diagnostic and treatment approaches (*Jull et al., 2007*).

*Mayer et al. (2012)* developed the Central Sensitization Inventory (CSI) as a patient-reported screening tool that can be used to identify and quantify CS/CSS-related symptomology. The CSI concept is based on the CSSs paradigm, in which distinct diseases

with different phenotypes share overlapping CS symptoms. Through a literature search, these symptoms were taken from the CSS paradigm and reduced into a single questionnaire (*Yunus, 2007*). The CSI is used to detect symptoms of CS, including widespread pain patterns, sleep disturbances, hypersensitivity to stimuli, and cognitive, digestive, and urological problems (*Mayer et al., 2012*). The CSI is a widely used tool that has demonstrated good reliability and validity in populations with chronic pain conditions (*Mayer et al., 2012*).

Recently, the CSI has received a lot of attention, and it has been translated, culturally adapted, and validated in numerous languages, including Brazilian Portuguese (*Caumo et al., 2017*), Dutch (*Kregel et al., 2016*), French (*Pitance et al., 2016*), Spanish (*Cuesta-Vargas et al., 2016*), Italian (*Chiarotto et al., 2018*), Serbian (*Knezevic et al., 2018*), and Japanese (*Tanaka et al., 2017*). While the CSI was also adapted and validated for use in Arabic-speaking populations, the study had several limitations (*Madi et al., 2022*). Firstly, it included participants with a wide range of chronic conditions, rather than focusing specifically on musculoskeletal pain, and it was conducted during the COVID-19 pandemic, which may have introduced variability that could affect the results. Secondly, the study did not assess potential floor and ceiling effects, which are crucial for understanding the responsiveness and interpretability of the CSI in this particular context. Thus, the aim of the current study was to translate and culturally adapt the CSI into the Arabic language (CSI-Ar), as well as to evaluate its test-retest reliability, construct validity, and discriminant validity in patients with chronic musculoskeletal pain disorders.

## METHODS

### Design
The design of this study was cross-sectional. After gaining permission of the author of the original English version, the process of translating and validating the CSI into Arabic was started. This study consisted of two stages: First, translation and cross-cultural adaptation of the original CSI version into Arabic; Second, assessing the psychometric properties of the Arabic version of the CSI. The study protocol received approval from the Scientific Research Ethics Committee at Taif University (No. 44-003). Written informed consent has been obtained from all participants. This study adhered to the guidelines set forth by "The Strengthening the Reporting of Observational Studies in Epidemiology (STROBE)" and "COnsensus-based Standards for the selection of health Measurement INstruments (COSMIN)" guidelines (*Von Elm et al., 2007*; *Gagnier et al., 2021*).

### Translation and cross-cultural adaptation of the CSI
This stage followed the criteria for adapting self-report measures for cross-cultural adaptation and translation (*Beaton et al., 2000*):

**Step 1: forward translation.** The CSI was translated from English to Arabic with the goal of preserving the original questionnaire's meaning. Two translations were completed by two translators who speak Arabic as their native language. The translators transferred the item to an appropriate cultural context when a concept had no equivalent in Arabic culture. A discussion between the two translators was held to determine the translational

options for the most difficult terms. The translators then worked on combining the two translations into a single translation. None of the original items were excluded.

**Step 2: backward translation**. Two independent bilingual native English-speaking translators worked on the back translation from the Arabic version into English while taking into account social and cultural differences between the US and Arab. To reduce information bias and allow unexpected interpretations of questions in the translated questionnaire, the two translators were not aware about the topics being studied, and they did not have medical backgrounds.

**Step 3: expert committee.** A multilingual committee, which included our four translators, reviewed both forward and backward translations. To achieve conceptual equivalency, the group discussed various choices for items and responses, emphasizing meaning above literal translation.

**Step 4: testing the pre-final version**. The questionnaire was given to 50 patients who were randomly chosen from all patients at the participating facilities who met the inclusion and exclusion criteria to assess the clarity of items and responses of the Arabic CSI (CSI-Ar) and on how to revise them if necessary.

## Participants

This study included adults with chronic musculoskeletal pain for a minimum of 3 months, and who have enough knowledge of the Arabic language and sufficient physical and cognitive ability to participate. This study excluded those with a diagnosis of specific medical conditions that can negatively impact the central nervous system, including brain or spinal cord injury, cancer and/or neurological disease or injury. It also excluded participants with psychiatric disease with pain as the main symptom (for example somatoform disorders, severe depression), as well as participants whose rheumatological disease was in its initial or unstable phase, and/or participants with a primarily neuropathic pain component.

In addition, local contacts were used to recruit healthy control participants, without musculoskeletal pain, from the general population.

## Sample size calculations

The sample size was estimated following the methodology outlined by *Boateng et al. (2018)*. It is suggested that a minimum of 10 participants per scale item is required, with an ideal ratio of 10:1. Because the CSI has 25 items, this study requires 250 participants. The aim was to recruit a minimum of 250 participants with chronic musculoskeletal pain and 50 healthy control participants.

## Administered questionnaires

### Central Sensitization Inventory-Arabic version

The CSI is composed of two parts (A and B). Part A has 25 questions that assess the typical symptoms of CS/CSS. The severity of these symptoms is rated on a five-point Likert scale, from never to always (never = 0, rarely = 1, sometimes = 2, often = 3, always = 4). The single-item scores are added up to provide a total score ranging from 0 to 100. Part B of the

questionnaire questions the patient about ten previously diagnosed disorders from their medical history, including seven common CSSs and three additional conditions associated with CS/CSS. Part B of the CSI is not assessed in the same way as Part A and is just intended to offer extra information (*Mayer et al., 2012*; *Neblett, 2018*). The authors have been granted permission from the copyright holders to translate and assess the psychometric properties of this instrument.

### Depression anxiety stress scale

The Depression, Anxiety, and Stress Scale (DASS) is a tool developed for evaluating and assessing the depression, anxiety, and stress (*Lovibond, 1995*). The authors have obtained permission from the copyright holder to use this instrument. The DASS-21 is a new short version of DASS which includes three subscales, each containing seven items. Many studies have assessed the psychometric prosperities of this scale to determine its validity and reliability (*Ali et al., 2017*; *Pezirkianidis et al., 2018*; *Yıldırım, Boysan & Kefeli, 2018*; *Jiang et al., 2020*; *Zanon et al., 2021*).

### Pain catastrophizing scale

The Pain Catastrophizing Scale (PCS) is used to quantify catastrophizing attitudes and beliefs about pain (*Sullivan, Bishop & Pivik, 1995*). It consists of 13 items, each of which is assessed on a five-point Likert scale, with a total score ranging from 0 to 52 (*Meyer, Sprott & Mannion, 2008*). The PCS has previously been translated, culturally adapted, and validated into Arabic (*Terkawi et al., 2017*). The authors have obtained permission from the copyright holder to use this instrument.

### EuroQOL's five-dimension questionnaire

The EQ-5D is a general, preference-based instrument that assesses three different aspects of quality of life (*Rabin & De Charro, 2001*). The first aspect is a descriptive system with a five-dimensional profile of respondents' health state. The second aspect is a visual analog scale (VAS; 0–100) for rating one's own health. The third aspect of the questionnaire is an index score that reflects the general public's choice or utility for the measured health profile can be created. Previously, the tool was translated and validated into Arabic (*Bekairy et al., 2018*). The authors have obtained permission from the copyright holder to use this instrument.

### Numeric pain rating scale

The Numeric Pain Rating Scale (NPRS) is an outcome measure that measures pain intensity in adults (*Jensen & McFarland, 1993*). Patients rate their pain using the 11-point numerical pain rating scale (NPRS), which has 11 different categories (*Downie et al., 1978*; *Jensen, Turner & Romano, 1994*; *Price et al., 1994*; *Katz & Melzack, 1999*). Furthermore, it has been demonstrated to have concurrent and predictive validity of pain intensity (*Jensen, Turner & Romano, 1994*; *Price et al., 1994*; *Jensen et al., 1999*; *Katz & Melzack, 1999*). The NPRS was previously translated into Arabic which was valid and reliable for measuring pain levels in patients with knee osteoarthritis (*Alghadir, Anwer & Iqbal, 2016*).

### Tampa scale of kinesiophobia

The original Tampa Scale of Kinesiophobia (TSK) was first developed in 1991 and it measures the subjective level of kinesiophobia or fear of movement (*Miller, Kori & Todd, 1991*). It comprises of 17 questions, each of which asks about the severity of the symptoms and discomfort. The responses are scored on a four-point Likert scale, where "totally disagree" equals one point, "partially disagree" equals two points, "partially agree" equals three points, and "totally agree" equals four points. To calculate the overall score, the answers to questions 4, 8, 12, and 16 must be inverted. The potential score on the scale can be anywhere between 17 and 68 points, with a higher number indicating a higher level of kinesiophobia. The Arabic version of the TSK was also found to be reliable and valid (*Yangui et al., 2017*).

## Data collection

An online-based survey (Google Forms survey) was created for participants to fill out. The participants had the choice to fill out the questionnaires in the hospital or at home. Social media was used to invite eligible patients to participate in this study. Participants were kindly instructed to complete all the parts of the questionnaire.

## Statistical analysis

### Descriptive analyses

All the analyses were conducted using the SPSS software (version 26.0, IBM Corp., Armonk, NY, USA). Means, standard deviations (SDs) and numbers were used to present the results of descriptive analyses. The normality of data was assessed by Kolmogorov-Smirnov test.

### Factor analysis

The data were deemed appropriate for factor analysis if the Bartlett's test of sphericity had a $p$-value less than 0.05 and the Kaiser-Meyer-Olkin measure of sampling adequacy exceeded 0.80 (*Kaiser, 1974*). An exploratory factor analysis was performed using maximum-likelihood extraction, or principal axis factoring if the data were not normally distributed (*Costello & Osborne, 2005*). Factors were extracted based on three criteria: the inflection point on the Scree plot (where factors above this point were retained), eigenvalues greater than 1.0, and each factor accounting for more than 10% of the variance (*Costello & Osborne, 2005*). Promax rotation with Kaiser normalization was utilized to estimate the loading of each item on the extracted factor. An item qualified for factor inclusion when it exhibited a factor loading coefficient greater than 0.3 (*Costello & Osborne, 2005*). McDonald's omega total (ωt) was estimated to confirm that the factors derived from the factor analysis were reliable (*Dunn, Baguley & Brunsden, 2014*). An omega value above 0.70 is considered acceptable (*McDonald, 2013*).

### Floor and ceiling effects

To determine whether there are any floor and ceiling effects, the distribution of the CSI-Ar score was examined. The lowest possible score on the CSI-Ar is (0 = floor), while the highest possible score is (100 = ceiling). Reliability and validity of a scale can be

compromised if the percentage of participants with the lowest and highest scores exceeded 15% (*Terwee et al., 2007*).

### Internal consistency and test-retest reliability

Cronbach's alpha test has been utilized to assess the internal consistency of the CSI-Ar items. A value range of 0.70 to 0.95 for Cronbach's alpha indicates acceptable internal consistency (*Terwee et al., 2007*). For analysis of test-retest reliability (consistency and stability of the CSI-Ar tool over time where no changes expected), a 2-week interval between two measures was required. A previous study recommended the test-retest time interval of 2 weeks which was long enough to prevent participants from forgetting previous answers but short enough to avoid changes in health conditions affecting replies (*De Vet et al., 2011*).

For the assessment of test-retest reliability of the CSI-Ar, the two-way random intraclass correlation coefficients (ICCs) for absolute agreement was used, along with the corresponding 95% confidence intervals (95% CIs) (*Terwee et al., 2007*). The reliability was rated as "excellent" ($ICC \geq 0.75$), "good" ($0.40 \leq ICC < 0.75$), or "poor" ($ICC < 0.40$). The standard error of measurement (SEM) was computed using the formula: $SEM = SD_{pooled\ standard\ deviation} \times \sqrt{(1 - ICC)}$. The smallest detectable change for the individual score ($SDC_{individual}$) was computed using the following formula: $SDC_{individual} = 1.96 \times \sqrt{2} \times SEM$. Then, the smallest detectable change for the group score ($SDC_{group}$) was computed using the following formula: $SDC_{group} = SDC_{individual}/\sqrt{n}$, where n means number of participants (*Terwee et al., 2007*).

### Construct validity

To assess to which degree the CSI-Ar tool is assessing the concept it claims to measure, the construct validity was assessed. The Spearman's rho correlation was used to assess the construct validity between total CSI-Ar scores and scores of the PCS, DASS-21, TSK, EQ-5D, and EuroQol VAS. The correlation was classified as "weak" (Spearman's rho < 0.3), "medium" ($0.3 \leq$ Spearman's rho < 0.5) or "strong" (Spearman's rho $\geq 0.5$) (*Cohen, 2016*).

### Discriminant validity

To ensure that the CSI-Ar tool is measuring the specific concept it claims to measure and not overlapping with unrelated constructs, the discriminant validity was assessed. The Separate Mann-Whitney U-tests were performed to test the discriminant validity of the total CSI-Ar score between 1) patients' group and healthy control group, 2) patients with at least one physician-diagnosed disorder in Part B of the CSI-Ar and patients without physician-diagnosed disorders, and 3) patients with self-reported single site pain and patients with self-reported multisite pain (pain in two or more sites).

## RESULTS

The CSI was forward and backward translated into Arabic without any major obstacles. In the pre-testing phase, no comprehensibility issues emerged, as the participants reported that items of the CSI–Ar were clear and easy to understand. Therefore, no changes were made to the CSI–Ar version after the pretest phase. The results of the

Kolmogorov-Smirnov tests indicated that none of the outcome measures followed a normal distribution ($P < 0.01$).

## Participants' characteristics

Table 1 presents the demographic characteristics of participants with different musculoskeletal pain disorders ($n = 264$) and healthy controls ($n = 56$). The average scores of the CSI-Ar (Part A) were $31.59 \pm 16.69$ and $12.88 \pm 12.51$ for the patients and healthy controls, respectively. In Part B of the CSI-Ar, 76 patients (28.79%) reported at least one physician-diagnosed disorder, while the remaining patients did not report any diagnosis.

## Factor analysis

The results of the KMO and Bartlett's test indicate that the data are suitable for factor analysis, with a KMO of 0.915 and a significant Bartlett's test of sphericity ($P < 0.001$). The principal axis factoring revealed five possible eigenvalues greater than 1.0. The first eigenvalue (8.58) accounted for 32.4% of the total variance, whereas the second to the fifth eigenvalues (ranged between 1.73 and 1.09) accounted for <10% of the variance. Visual inspection of the Scree plot (Fig. 1) showed inflection point at the second eigenvalue, which indicated a 1-factor model. After re-running the analysis to extract a single factor, this factor accounted for 31.9% of the total variance. As shown in Table 2, two items (items 10 and 18) were not loaded in the factor matrix. Removing these items resulted in a one-factor solution comprising 23 items, explaining 34.2% of the variance. The unidimensionality of the CSI-Ar (part A) was confirmed by the high value of McDonald's omega total ($\omega t = 0.92$). The subsequent measurement properties were conducted using this established factor structure. The total scores of the 23-item CSI-Ar ranges from 0–92.

## Floor and ceiling effects

No floor and ceiling effects were found. None of the patients scored the lowest (sum score = 0) or highest (sum score = 100) score on the CSI-Ar scale. Of the 264 patients, only one patient (0.4%) had the lowest CSI-Ar score, whereas none of the patients had the highest CSI-Ar score.

## Internal consistency

The internal consistency of the CSI-Ar was considered to be acceptable whereas the Cronbach's alpha value was 0.919.

## Test-retest reliability

The CSI-Ar was completed twice by 47 patients, with a 2-week delay between the two assessments. The CSI-Ar's test-retest reliability was excellent, with an $ICC_{2,1}$ of 0.874 (95% CI [0.785–0.928], $P < 0.001$). The SEM was 5.80, the $SDC_{individual}$ was 16.09 and the $SDC_{group}$ was 2.23. The average scores of the CSI-Ar were $27.64 \pm 16.58$ at the first test and $27.23 \pm 16.14$ at the second test. The Wilcoxon Signed Rank test showed no significant differences between the first and second average scores of the CSI-Ar ($P = 0.86$).

**Table 1 Demographic and clinical characteristics of participants.**

|  | Patients with different musculoskeletal pain disorders (n = 264) | Healthy controls (n = 56) |
|---|---|---|
| Age, year | 42.66 ± 15.33 | 33.23 ± 15.86 |
| Sex n |  |  |
| Women | 175 | 38 |
| Men | 89 | 18 |
| Height, cm | 161.44 ± 10.41 | 160.54 ± 10.11 |
| Weight, kg | 77.16 ± 17.84 | 63.91 ± 17.16 |
| Education level, n |  |  |
| Illiterate | 24 | 0 |
| Elementary | 17 | 2 |
| Middle | 18 | 2 |
| High | 37 | 9 |
| Bachelor's degree | 147 | 35 |
| Master's degree | 18 | 8 |
| PhD's degree | 3 | 0 |
| Smoking status, n |  |  |
| Current smoker | 59 | 16 |
| Former smoker | 29 | 3 |
| Never smoked | 176 | 37 |
| Marital status |  |  |
| Married | 139 | 19 |
| Single | 75 | 31 |
| Divorced | 32 | 6 |
| Widowed | 18 | 0 |
| Number of pain sites, n |  |  |
| 0 | 6 | – |
| 1 | 143 | – |
| 2 | 53 | – |
| 3 | 31 | – |
| 4 | 12 | – |
| 5 or more | 19 | – |
| Most referred areas, n |  |  |
| Neck | 32 | – |
| Shoulders | 31 | – |
| Upper back | 23 | – |
| Low back | 35 | – |
| Elbow | 7 | – |
| Wrists/hands | 16 | – |
| Hips/thighs | 23 | – |
| Knees | 33 | – |
| Ankles/feet | 24 | – |
| CSI-Ar, Part A | 27.97 ± 16.08 | 11.27 ± 11.50 |

(Continued)

| Table 1 (continued) | | |
|---|---|---|
| | Patients with different musculoskeletal pain disorders ($n$ = 264) | Healthy controls ($n$ = 56) |
| CSI-Ar, Part B, $n$ | | |
| Restless leg syndrome | 6 | 0 |
| Chronic fatigue syndrome | 10 | 0 |
| Fibromyalgia | 4 | 0 |
| Temporomandibular joint disorder | 2 | 1 |
| Migraine or tension headaches | 23 | 5 |
| Irritable bowel syndrome | 45 | 9 |
| Multiple chemical sensitivities | 4 | 2 |
| Neck injury (including whiplash) | 5 | 0 |
| Anxiety or panic attacks | 18 | 1 |
| Depression | 21 | 3 |
| NPRS | 6.40 ± 2.26 | – |
| Pain catastrophizing scale | | |
| Rumination | 6.48 ± 3.93 | – |
| Magnification | 3.67 ± 3.25 | – |
| Helplessness | 7.64 ± 6.15 | – |
| Total score | 17.79 ± 12.19 | – |
| Depression, Anxiety and stress scale | | – |
| Depression | 5.33 ± 4.79 | – |
| Anxiety | 4.10 ± 4.43 | – |
| Stress | 6.69 ± 4.69 | – |
| Total score | 16.12 ± 12.72 | – |
| Tampa scale of Kinesiophobia | 37.17 ± 8.32 | – |
| 5-level EuroQol-5D | 0.67 ± 0.27 | – |
| EuroQol VAS | 63.29 ± 29.43 | – |

**Notes:**
Data are presented as mean ± SD. Abbreviations: CSI-Ar, Arabic version of the Central Sensitization Inventory, NPRS, Numeric pain rating scale.

## Construct validity

The results showed that the CSI-Ar scale had significant correlations with all PCS subscales and total score (Spearman's rho = 0.459–0.563, $P < 0.001$), all DASS-21 subscales and total score (Spearman's rho = 0.599–0.685, $P < 0.001$), the TSK (Spearman's rho = 0.395, $P < 0.001$), and the EQ-5D (Spearman's rho = −0.396, $P < 0.001$). The results of the correlation analyses are demonstrated in detail in Table 3.
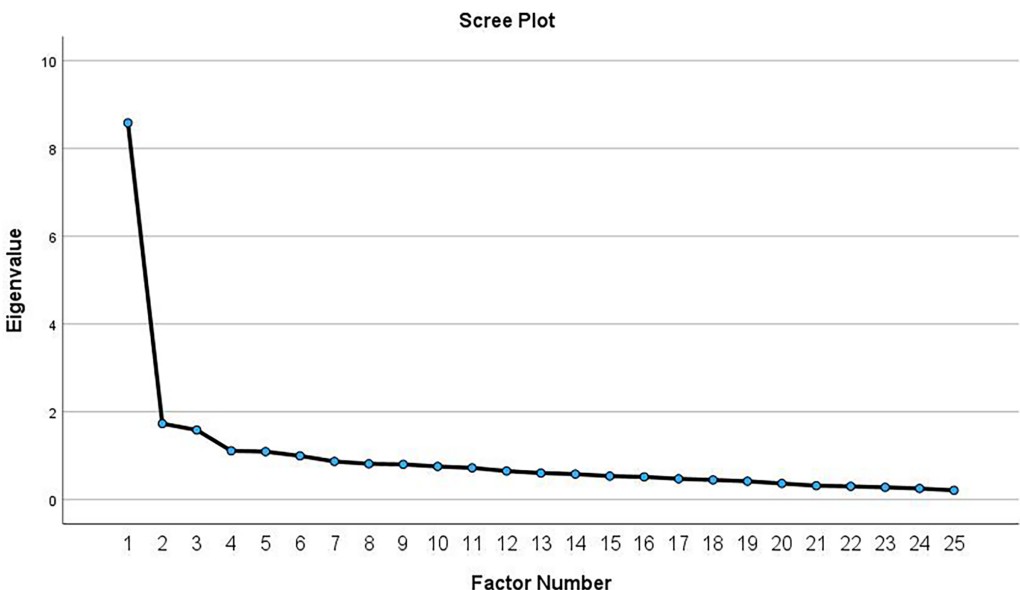

**Figure 1 Scree plot from the exploratory factor analysis of the CSI-Ar among patients with chronic musculoskeletal pain.**

**Table 2 Factor loadings extracted using principal axis factor and promax rotation with Kaiser normalization.**

| Individual items | Factor 1 |
|---|---|
| 1. I feel tired and unrefreshed when I wake from sleeping. | 0.619 |
| 2. My muscles feel stiff and achy. | 0.416 |
| 3. I have anxiety attacks. | 0.515 |
| 4. I grind or clench my teeth. | 0.335 |
| 5. I have problems with diarrhea and/or constipation. | 0.551 |
| 6. I need help in performing my daily activities. | 0.597 |
| 7. I am sensitive to bright lights. | 0.534 |
| 8. I get tired very easily when I am physically active. | 0.66 |
| 9. I feel pain all over my body. | 0.727 |
| **10. I have headaches.** | |
| 11. I feel discomfort in my bladder and/or burning when I urinate. | 0.637 |
| 12. I do not sleep well. | 0.555 |
| 13. I have difficulty concentrating. | 0.745 |
| 14. I have skin problems such as dryness, itchiness, or rashes. | 0.468 |
| 15. Stress makes my physical symptoms get worse. | 0.665 |
| 16. I feel sad or depressed. | 0.632 |
| 17. I have low energy. | 0.782 |
| **18. I have muscle tension in my neck and shoulders.** | |
| 19. I have pain in my jaw. | 0.321 |
| 20. Certain smells, such as perfumes, make me feel dizzy and nauseated. | 0.439 |
| 21. I have to urinate frequently. | 0.684 |

(Continued)

| Individual items | Factor 1 |
|---|---|
| 22. My legs feel uncomfortable and restless when I am trying to go to sleep at night. | 0.662 |
| 23. I have difficulty remembering things. | 0.678 |
| 24. I suffered trauma as a child. | 0.365 |
| 25. I have pain in my pelvic area. | 0.533 |

Note:
    *Bolded items were not loaded on the factor matrix.

**Table 3 Correlations between the CSI-Ar scale and other outcome measures in patients with different musculoskeletal pain disorders ($n$ = 264).**

| | CSI-Ar scale | |
|---|---|---|
| | Spearman's rho | $P$ |
| NPRS | 0.282 | <0.001 |
| Pain catastrophizing scale | | |
| Rumination | 0.469 | <0.001 |
| Magnification | 0.459 | <0.001 |
| Helplessness | 0.563 | <0.001 |
| Total score | 0.552 | <0.001 |
| Depression, Anxiety and Stress Scale (DASS-21) | | |
| Depression | 0.625 | <0.001 |
| Anxiety | 0.655 | <0.001 |
| Stress | 0.599 | <0.001 |
| Total score | 0.685 | <0.001 |
| Tampa Scale of Kinesiophobia | 0.395 | <0.001 |
| 5-level EuroQol-5D | −0.396 | <0.001 |
| EuroQol VAS | −0.222 | <0.001 |

Note:
    Abbreviations: CSI-Ar, Arabic version of the Central Sensitization Inventory; NPRS, numeric pain rating scale;
    VAS, Visual analog scale.

## Discriminative validity

The Mann-Whitney U-test showed a statistically significant difference between the patient group and the healthy control group ($P < 0.001$), with the healthy controls displaying a lower average CSI-Ar score (12.27 ± 11.50) when compared to the patient group (27.97 ± 16.08) (Table 1).

When compared to patients with at least one physician-diagnosed disorder in Part B of the CSI-Ar ($n$ = 76), the Mann-Whitney U-test revealed that patients without physician-diagnosed disorders ($n$ = 188) had a significantly lower average CSI-Ar score (36.97 ± 17.11 *vs.* 24.33 ± 14.14, respectively, $P < 0.001$). Further, the patients with self-reported single-site pain ($n$ = 143) had a statistically significantly lower CSI-Ar score when compared to patients with self-reported multisite pain ($n$ = 115) (25.04 ± 15.23 *vs.* 31.29 ± 15.99, $P = 0.001$).

## DISCUSSION

This study translated, culturally adapted, and validated the CSI-Ar version in patients with different chronic musculoskeletal pain disorders. The results showed that the CSI-Ar has satisfactory psychometric properties, when compared to those of the original English version (*Mayer et al., 2012*).

The structural validity of the CSI has shown variability across different languages (*Pitance et al., 2016*; *Cuesta-Vargas et al., 2016*; *Tanaka et al., 2017*; *Chiarotto et al., 2018*; *Knezevic et al., 2018*; *van der Noord, Paap & van Wilgen, 2018*). However, a recent multi-national study suggested that only total CSI scores should be used and reported (*Cuesta-Vargas et al., 2018*), confirming its unidimensionality and validating the use of the total score. Our research aligns with this study, supporting the unidimensionality of the CSI-Ar. Our factor analysis revealed a one-factor solution that accounted for a significant proportion of variance, providing evidence of structural validity. The results of the current study are similar to other adaption studies such as Italian (*Chiarotto et al., 2018*) and Spanish (*Cuesta-Vargas et al., 2016*). However, these findings differ from the English (*Mayer et al., 2012*), Dutch (*Kregel et al., 2016*), and Brazilian Portuguese (*Caumo et al., 2017*) versions, which identified more than one factor.

The internal consistency of the CSI-Ar was considered to be acceptable with a Cronbach's alpha value of 0.915. This finding has been found to be consistent with the values of the internal consistency of the other versions of the CSI scale of other languages which had Cronbach's alpha values ranging from 0.87 (*Chiarotto et al., 2018*; *Sharma et al., 2020*) to 0.993 (*Bilika et al., 2020*). Furthermore, the CSI-Ar's test-retest reliability was excellent, with an ICC of 0.872, which is comparable to the results of previous studies that had test-retest reliability values ranging from 0.85 (*Tanaka et al., 2017*a) to 0.991(*Bilika et al., 2020*). The SEM in our study was 6.19 and the SDC was 2.38, whereas the SEM values of other studies ranged from 0.31 (*Sharma et al., 2020*) to 3.16 (*Knezevic et al., 2018*), and the SDC ranged from 0.86 (*Sharma et al., 2020*) to 8.12 (*Knezevic et al., 2018*). One possible explanation for the higher SEM in our study could be related to the characteristics of our study sample. It is important to consider the demographics and clinical characteristics of the participants in our study, as these factors can influence the variability of responses and ultimately impact the SEM. Additionally, differences in the administration of the CSI-Ar, such as variations in the instructions given to participants or the setting in which the assessment took place, could contribute to the variation in SEM values across studies.

Assessing construct validity, the results showed that the CSI-Ar scale had positive medium to strong correlations with all PCS subscales. Previous versions of the scale of the other languages reported also a significant correlation between the PCS subscales and CSI (*Caumo et al., 2017*; *Bilika et al., 2020*). Moreover, the CSI-Ar scale showed a positive and strong correlation with the DASS-21 subscales and the total score, which is not in line with the German CSI version (*Klute et al., 2021*), as they reported that all the three scales of the DASS-21 showed a low, positive correlation with the German CSI version, indicating an increasing negative emotional stress with increasing scores. This discrepancy between the Arabic and German versions of the CSI suggests that the relationship between the CSI and

DASS-21 scales may be influenced by cultural and contextual factors. Further research is needed to understand these differences and their implications for the use of the CSI in different cultural settings.

The results showed a significant difference between the patient group and the healthy control group, with the healthy controls displaying a lower average CSI-Ar score compared to the healthy control group. Moreover, when compared to patients with at least one physician-diagnosed disorder in Part B of the CSI-Ar, the results revealed that patients without physician-diagnosed disorders had a significantly lower average CSI-Ar score. Furthermore, patients with self-reported single-site pain had a significantly lower CSI-Ar score when compared to patients with self-reported multisite pain. The results were expected since each of these groups had a different level of CS.

One limitation of the current study is that Saudi Arabia is a multicultural country, and its residents speak different accents, which may influence some concepts and aspects of the CSI-Ar. It is recommended to conduct further validation studies of the CSI-Ar in larger and more diverse populations, including individuals from different regions and cultural backgrounds within Saudi Arabia, to account for potential linguistic and cultural variations. Another limitation is that women outnumbered men in this study, which may affect the concepts of pain and, consequently, the study results. It is recommended to conduct further studies investigating the influence of gender differences on the interpretation and response patterns to the CSI-Ar.

## CONCLUSION

The CSI-Ar was found to be internally consistent, reliable, and valid in Arabic-speaking participants who complained of chronic musculoskeletal pain. The results showed excellent test-retest reliability and acceptable internal consistency. Moreover, the results had found that the psychometric properties of the CSI-Ar corresponded to the original English version and the other previous CSI versions of various languages. The CSI-Ar can be a useful screening tool for the clinicians and researchers to assess the CS/CSS-related symptoms in participants with chronic musculoskeletal pain. Future research should explore the responsiveness of the CSI-Ar in detecting changes in central sensitization over time or in response to treatment, to establish its utility for monitoring and guiding interventions. Additionally, it is recommended to assess the predictive validity of the CSI-Ar in identifying subgroups of patients who may benefit from targeted interventions for chronic musculoskeletal pain.

### Funding

This research was funded by Taif University, Taif, Saudi Arabia (TU-DSPP-2024-177). The funders had no role in study design, data collection and analysis, decision to publish, or preparation of the manuscript.

## Grant Disclosures

The following grant information was disclosed by the authors:

Taif University, Taif, Saudi Arabia: TU-DSPP-2024-177.

## Competing Interests

The authors declare that they have no competing interests.

## Author Contributions

- Sarah E. Tamboosi conceived and designed the experiments, performed the experiments, prepared figures and/or tables, authored or reviewed drafts of the article, and approved the final draft.
- Hosam Alzahrani conceived and designed the experiments, performed the experiments, analyzed the data, prepared figures and/or tables, authored or reviewed drafts of the article, and approved the final draft.
- Fahad H. Alshehri performed the experiments, authored or reviewed drafts of the article, and approved the final draft.
- Msaad Alzhrani performed the experiments, authored or reviewed drafts of the article, and approved the final draft.
- Yasir S. Alshehri analyzed the data, prepared figures and/or tables, authored or reviewed drafts of the article, and approved the final draft.

## Human Ethics

The following information was supplied relating to ethical approvals (*i.e.*, approving body and any reference numbers):

The study protocol received approval from the Scientific Research Ethics Committee at Taif University (No. 44-003).

## Data Availability

The raw measurements are available in the Supplemental File.

## Supplemental Information

Supplemental information for this article can be found online at http://dx.doi.org/10.7717/peerj.18251#supplemental-information.

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
