# Peer review of "Cross-cultural adaptation and psychometric properties of the Arabic version of the Central Sensitization Inventory in people with chronic musculoskeletal pain"

_PeerJ, doi:10.7717/peerj.18251_

## Round 0.1 · original submission · Major Revisions

The reviewers found merit in your manuscript but have requested additional details with your statistical analysis and results.

Reviewer 1 ·

Basic reporting

no comment

Experimental design

no comment

Validity of the findings

no comment

·

Basic reporting

This is an observational cross-sectional study aimed at translating, culturally adapting, and validating the Central Sensitization Inventory for the Arabic-speaking population with chronic musculoskeletal pain disorders. The literature that has been presented fulfills a solid background about the manuscript's subjects. The authors utilized a suitable and valid methodology in the aspect of the translation and the validation process and depicted the results efficiently. Finally, they discussed and compared their results with the previous validation studies of the other languages.

- However, some points will improve the manuscript dramatically as stated below. In addition, running the factor analysis including the expletory (EFA) and confirmatory (CFA) will ultimate the psychometric properties because of the large sample and its suitability.

The suggestion to improve the manuscript format and structure, read and follow the undermentioned guidelines:
o The STROBE guideline to adhere to the format of the paper.
 Cuschieri S. The STROBE guidelines. Saudi J Anaesth. 2019 Apr;13(Suppl 1):S31-S34. doi: 10.4103/sja.SJA_543_18. PMID: 30930717; PMCID: PMC6398292.

o The COSMIN methodology is under the following references:
 Mokkink LB, de Vet HCW, Prinsen CAC, Patrick DL, Alonso J, Bouter LM, Terwee CB. COSMIN Risk of Bias checklist for systematic reviews of Patient-Reported Outcome Measures. Qual Life Res. 2018 May;27(5):1171-1179. doi: 10.1007/s11136-017-1765-4. Epub 2017 Dec 19. PMID: 29260445; PMCID: PMC5891552.
 Prinsen CAC, Mokkink LB, Bouter LM, Alonso J, Patrick DL, de Vet HCW, Terwee CB. COSMIN guideline for systematic reviews of patient-reported outcome measures. Qual Life Res. 2018 May;27(5):1147-1157. doi: 10.1007/s11136-018-1798-3. Epub 2018 Feb 12. PMID: 29435801; PMCID: PMC5891568.
 Terwee CB, Prinsen CAC, Chiarotto A, Westerman MJ, Patrick DL, Alonso J, Bouter LM, de Vet HCW, Mokkink LB. COSMIN methodology for evaluating the content validity of patient-reported outcome measures: a Delphi study. Qual Life Res. 2018 May;27(5):1159-1170. doi: 10.1007/s11136-018-1829-0. Epub 2018 Mar 17. PMID: 29550964; PMCID: PMC5891557.

• The language is clear and flows smoothly through the manuscript. However, exchange all seven sentences that hold the verb “we” into a passive voice.

• The literature references were sufficient and fulfilled the background.

• The article structure, figures, and tables were presented professionally. However, I recommend following the abovementioned guidelines (STROBE and COSMIN) to support the manuscript.

• The research question was defined well, relevant, meaningful, and filled an identified knowledge gap.

Experimental design

Material and methods
- Line 162: under the “2.5.1 Central Sensitization Inventory-Arabic version” section.
o State the psychometric properties of the original version to confirm its validity and reliability.

- Line 239-242: The authors stated in the manuscript that “A previous study recommended the test-retest time interval of 2 weeks which was long enough to avoid participants from forgetting previous answers but short enough to avoid changes in health conditions affecting replies.”.
o What study was stating this?

- Line 260-265: The authors have chosen the “Separate Mann-Whitney U-tests were performed to test the discriminant validity of the total CSI-Ar score between ...”.
o Why do you go for a non-parametric test rather than a parametric test? Justify this with the normality tests!
o What are the expected results available here?
o Could you support your choice with a reference?

Validity of the findings

Results
- Line 269-270: “… as the participants reported that items of the CSI-Ar were clear and easy to understand.”
o Consider the face and content validity within these results.
o Read Terwee CB, Prinsen CAC, Chiarotto A, Westerman MJ, Patrick DL, Alonso J, Bouter LM, de Vet HCW, Mokkink LB. COSMIN methodology for evaluating the content validity of patient-reported outcome measures: a Delphi study. Qual Life Res. 2018 May;27(5):1159-1170. doi: 10.1007/s11136-018-1829-0. Epub 2018 Mar 17. PMID: 29550964; PMCID: PMC5891557.

- Line 290-291: “The CSI-Ar was completed twice by 47 patients, with a 2-week delay between the two assessments.”
o Provide from the literature a reference

Additional comments

Conclusions
- State the recommendations from this study for the researchers and readers.

·

Basic reporting

I reviewed the manuscript entitled “Cross-cultural adaptation and psychometric properties of the Arabic version of the Central Sensitization Inventory in people with chronic musculoskeletal pain ” The manuscript reported results of a cross-sectional study examining the cross-cultural adaptation and psychometric properties of the CSI-Ar amongst a sample of Arab patients with chronic musculoskeletal pain. The study was carried out in 2 phases. The first phase involved translation and cross-cultural adaptation of the PSEQ into the Arabic language. In the second phase, the reliability and construct validity of the PSEQ-A were people with chronic musculoskeletal pain. the manuscript suffers from some limitations and weak points which will be addressed as follows:
1. Introduction, last para, please include the measurement properties aimed to be evaluated.
2. Methods ,Was there any order for completing the questionnaires?
3. In methods, please define the psychometric properties that were evaluated and provide the reference on how to measure each type of validity.
4. In the test retest reliability section (Page 11, Lines 246 to 249) please provide a reference for the justification that no tool such as The Global Rating of Change was used to measure any changes in participants between the test and retest period “ A 2-week interval between two measures was required. A previous study recommended the test-retest time interval of 2 weeks which was long enough to avoid participants from forgetting previous answers but short enough to avoid changes in health conditions affecting replies ”.
5. As Terwee et al. (2007) mentioned, construct validity should be assessed by testing predefined hypotheses. High construct validity can be achieved by meeting 75% of the results that correspond with these hypotheses.
Consequently, there is doubt about the method that measured this type of validity because the author did not mention any prior hypotheses.
6. Exploratory factor analysis should be performed and reported for the CSI-Ar.
7. Please comment on the appropriateness of the sample size for the retest.
8. In the Discussion section, the standard error of measurement was 6.19, which is higher than the highest value of 3.16 found in other studies. Please provide justification for why the standard error of measurement for CSI-AE is twice as high as in other studies.
9. In the Discussion section (Page 15), please provide a justification for the difference between the CSI-Ar and the German CSI version correlation with the Depression Anxiety Stress Scale.
10. In the Discussion section (Page 15, Lines 349 to 353), the author replicates without any reason how to assess discriminant validity, as he previously mentioned in the method section (page 12).

Experimental design

No comment

Validity of the findings

No comment

---

## Round 0.2 · Major Revisions

The reviewer has requested additional justification/edits with the analysis.

·

Basic reporting

clear

Experimental design

Please see the PDF file

Validity of the findings

clear

Additional comments

see the PDF file

---

## Round 0.3 · Major Revisions

One reviewer required additional clarification for your statistical analysis and subsequent results.

---

## Round 0.4 · accepted · Accept

The authors addressed the reviewers concerns. The manuscript is ready for publication.

·

Basic reporting

accepted

Experimental design

accepted

Validity of the findings

accepted

Additional comments

non